# Accelerometer-Measured Physical Activity and Sedentary Behavior of Adults with Prader-Willi Syndrome Attending and Not Attending a Small-Scale Community Workshop

**DOI:** 10.3390/ijerph19159013

**Published:** 2022-07-25

**Authors:** Ming-Ju Wu, Li-Ping Tsai, Ting-Fu Lai, Jeong Su Cho, Yung Liao

**Affiliations:** 1Lifestyles of Health and Sustainability Executive Master of Business Administration, National Taiwan Normal University, Taipei 106, Taiwan; mandywu0912@gmail.com; 2Department of Otolaryngology, National Taiwan University Hospital Bei-Hu Branch, Taipei 108, Taiwan; 3Department of Pediatrics, Taipei Tzu Chi Hospital, Buddhist Tzu Chi Medical Foundation, New Taipei City 231, Taiwan; tsaiped@gmail.com; 4Department of Medicine, Tzu Chi University, Hualien 970, Taiwan; 5Department of Health Promotion and Health Education, National Taiwan Normal University, Taipei 106, Taiwan; ted971345@gmail.com; 6Department of Thoracic and Cardiovascular Surgery, Pusan National University School of Medicine, Medical Research Institute, Pusan National University Hospital, Busan 49241, Korea; 7Graduate Institute of Sport, Leisure and Hospitality Management, National Taiwan Normal University, Taipei 106, Taiwan

**Keywords:** health promotion, accelerometer, Prader-Willi syndrome

## Abstract

This cross-sectional study aimed to compare the accelerometer-assessed physical activity (PA) and sedentary behavior (SB) of adults with Prader-Willi syndrome (PWS) attending or not attending a small-scale community workshop (SSCW). A total of 18 adults with PWS were recruited in this study. Of these participants, 10 regularly attended an SSCW and 8 did not. All of the participants were asked to wear accelerometers for eight continuous days for measuring their PA and SB. The independent sample *t*-test was used. The results showed that the adults with PWS who attended the SSCW engaged in more moderate-to-vigorous PA (MVPA) and daily steps than those who did not. By stratifying between daytime/nighttime on weekdays, we found the participants who attended the SSCW had higher total PA, MVPA, daily steps, as well as lower total sedentary time, during the daytime on weekdays than those who did not. Policies or programs promoting PA and reducing SB among adults with PWS should thus consider providing structured programs or courses in a community center.

## 1. Introduction

Prader-Willi syndrome (PWS), first described in 1956 [1], is a chromosomal disorder with an estimated prevalence of 1/10,000–1/30,000 [2]. This congenital disease results from the absence of expression of paternal genes from chromosome 15q11.2-q13 [3]. There are three primary mechanisms underlying this lack of expression: Most PWS patients (65–75% of cases) have a deletion of a 5–6 Mb region from the paternally contributed chromosome 15; approximately 20–30% of PWS patients have maternal uniparental disomy (UPD); and 1–3% of PWS patients have an imprinting defect (ID) in the genomic region [4]. PWS is characterized by hypotonia, difficulty in feeding, failure-to-thrive in infancy, and then excessive weight gain after early infancy [5]. Delayed motor, cognitive, and language development are noted in children with PWS, and adults with PWS have mild intellectual disabilities and learning disabilities [6]. 

Overeating is the feature PWS that many people are most familiar with [7]. Young children with PWS seldom have this problem, but older children (most frequently, those older than 8 years old) and adults with PWS will start to have excessive appetites, engage in overeating and food seeking, and may be unable to achieve a sense of satiety [8]. As a result, obesity-related complications are the main causes of morbidity and mortality among PWS patients [9]. In addition to having their access to food restricted and receiving dietary behavior management, engaging in physical activity (PA) is a basic and important method of weight control for PWS patients [10]. However, young people and adults with PWS might have different opportunities to engage in PA in their daily routines and activities. For example, most young people with PWS go to school on weekdays, so they can engage in structured PA on weekdays. However, adults with PWS who less have the opportunity to engage in structured PA at school or who are not required to engage in PA by their guardians are less likely to obtain sufficient levels of autonomous PA [11]. This could be the reason why previous study from the U.S. reported that no adults with PWS met the recommended levels of PA [12]. Besides that, it is also worth noting that sedentary behavior (SB) is an any posture of low energy expenditure behavior [13] that may negatively influence an individual’s energy balance [14]. Although, studies have shown that higher levels of SB are associated with increased risks of obesity in adults [15,16]; however, few studies have explained the sedentary time and sedentary behavior patterns, such as duration and times of sedentary bout ≥30 min, among adults with PWS. Such information would be of value, however, in designing effective health-enhancing programs to help adults with PWS to avoid excessive SB. Relatedly, it is worthwhile to investigate whether adults with PWS who regularly attend structured programs or courses engage in higher levels of PA and avoid prolonged sedentary time in their daily lives.

Taiwan may offer a unique research opportunity in this context. In 2015 and 2018, the Prader-Willi Syndrome Association of Taiwan developed two small-scale community workshops (SSCWs) for PWS adults in northern and southern Taiwan. Each SSCW provides a place during the daytime for adult patients with PWS (aged ≥18 years) who have moderate self-care ability and require moderate support but have no school to go to, no job to do, or no family members available to take care of them after they have graduated from the compulsory education of high school. The SSCWs provide safe and learning-friendly environments in which these patients with PWS can engage in activities to help them maintain their physical function, prevent the exacerbation of the disease, improve their working ability, and learn skills in order to lead a more independent life. Adults with PWS can stay at the SSCWs during the daytime on weekdays and take part in scheduled classes, including exercise, physical therapy, and arts and crafts classes, all of which provide training in various skills. Theoretically, the given SSCW will arrange the daily weekday schedules of the participating adults with PWS to include an adequate amount of PA, such that the adults with PWS who attend an SSCW should likely engage in more PA and less SB than those who do not attend an SSCW. To strengthen the evidence used to test the hypothesis, this study used accelerometers to compare the time spent in PA (intensity-specific, daily steps) and SB (total sedentary time, sedentary bout times and duration) for adults with PWS who attended or did not attend an SSCW over a 7-day period, including during the daytime and nighttime on weekdays and a weekend.

## 2. Materials and Methods

### 2.1. Participants

As of May 2020, there were a total of 324 PWS patients in Taiwan, of whom 107 were adults (≥18 years old). Among these adults with PWS, a total of 44 were living in northern Taiwan and were the target population for our study. This study is a cross-sectional design. With regard to the recruitment of participants, we first explained the purposes and procedure of our study to social workers working with the PWS association in Taiwan. Then, those social workers helped us to recruit participants by making phone calls to the parents/guardians of the adults with PWS living in northern Taiwan (*n* = 44). Finally, eighteen adults with PWS agreed to take part in this study (participation rate: 40.9%), including ten participants who have attended the same SSCW for variable duration of 6 months to 5 years (every weekday), and eight participants who were not attending an SSCW. These eighteen participants were all confirmed to have received a diagnosis of PWS from medical centers in Taiwan. This study was approved by the Institutional Review Board of National Taiwan Normal University (201910HM006). All of the participants and their parents/guardians were informed of the nature of the study and signed informed consent forms prior to their participation.

### 2.2. Measures

#### 2.2.1. Physical Activity and Sedentary Behavior Measurement

The participants, their parents/guardians, and the social workers of the SSCW were instructed that each participant was to wear the GT3X+ accelerometer provided on an elastic belt placed across the waist line with the accelerometer itself over the left hip except when showering or swimming for eight consecutive days in order to obtain the accelerometer data of a whole week including five weekdays and a weekend. The accelerometers and daily activity forms were returned to the research team members on the final day of the study at the SSCW (by those participants attending the SSCW) or by mail (by those participants not attending the SSCW). The accelerometer data were downloaded and analyzed using the ActiLife software version 6.13.3 (ActiGraph, LLC, Pensacolsa, FL, USA).

The raw data processing included the extraction of the vertical axis data in 60-s epochs and the derivation of min of variable intensity of PA. Accelerometer counts ≥100 counts/min were defined as indicating PA; ≥100–2019 counts/min was defined as light-intensity physical activity (LPA); ≥2020 counts/min was defined as moderate-to-vigorous-intensity physical activity (MVPA). SB was confined to the non-sleeping time according to the daily activity forms reported by the participants and was defined with accelerometer counts <100 counts/min. Non-wearing time was defined as 60 min or more of continuous unbroken 0 counts with a tolerance of up to 2 min of limited movement. The total sedentary time, duration and times of sedentary bouts ≥ 30 min in length were calculated for the analysis. The drop time of a sedentary bout was set at two min when the data were analyzed [17]. A valid amount of accelerometer recording data for a day to be analyzed was defined as at least 10 h in a day. Participants with at least four valid weekdays and/or at least one weekend day of sufficient accelerometer data were included in this study [18]. Because the adults with PWS who attended the SSCW stayed at the SSCW from 9 am to 6 pm every weekday, in this study, the daytime was defined as the time from 9 am to 6 pm, and nighttime was defined as 6 pm until sleeping time.

#### 2.2.2. Anthropometrics

Each participant’s weight and height according to their parents or guardians were recorded to the nearest 1 kg and 1 cm, respectively. The body mass index (BMI) of each participant was then calculated as weight (kg)/square of height (m). The age and sex of each participant were also recorded.

### 2.3. Statistical Analyses

Descriptive statistics were calculated and shown as means with standard deviations or percentages of the subjects. The independent sample *t*-test was used to examine the significant differences in the mean ± standard deviation of the intensity-specific PA and SB patterns of the adults with PWS who attended and did not attend the SSCW during a 7-day period, the daytime and nighttime on the weekdays and on the weekend, separately. Statistical significance was indicated by a *p*-value of less than 0.05. All statistical analyses were performed with SPSS 23.0 (SPSS Inc., Chicago, IL, USA). 

## 3. Results

### 3.1. Description of Study Participants

Eighteen participants all provided valid data for analysis in this study. Table 1 shows the demographic variables of the two groups of participants (10 vs. 8), that is, the 10 adults with PWS who attended an SSCW and the 8 adults with PWS who did not attend an SSCW. Independent sample *t*-testing showed that the only characteristic of these two groups of participants that was significantly different was their age. Specifically, the mean age of the adults with PWS attending an SSCW was higher than that of the adults with PWS not attending an SSCW (28.4 ± 6.0 years vs. 22.6 ± 3.9 years, *p* = 0.032). The sex distributions of the two groups were no significant difference (*p* = 0.916). There were also no significant differences in height, weight, and BMI between the two groups.

### 3.2. Total Amounts and Patterns of Physical Activity and Sedentary Behavior in 7-Day Period

Table 2 shows the intensity-specific PA and SB patterns over a 7-day period. Within a 7-day period, the participants attending the SSCW spent more time engaged in MVPA (49.7 ± 29.6 vs. 22.6 ± 15.9, *p* = 0.033) and took more daily steps (9483.0 ± 3052.2 vs. 6260.4 ± 2733.4, *p* = 0.033) than did those not attending the SSCW. No significant differences between the groups in terms of total PA, LPA, or SB patterns were observed.

### 3.3. Total Amounts and Patterns of Physical Activity and Sedentary Behavior during the Daytime and Nighttime on Weekdays

In Table 3, during the daytime on weekdays, the participants attending the SSCW had more total PA (209.5 ± 31.9 vs. 167.8 ± 29.3, *p* = 0.012), MVPA (29.9 ± 18.2 vs. 13.1 ± 5.8, *p* = 0.024), and daily steps (6396.0 ± 2492.2 vs. 4047.6 ± 1650.2, *p* = 0.036) than the participants not attending the SSCW. Participants attending the SSCW also spent less total sedentary time (4.9 ± 0.5 vs. 5.7 ± 0.3, *p* = 0.002) compared with who did not attend SSCWs.

Table 3 also shows the intensity-specific PA and SB patterns during the nighttime on weekdays. During the nighttime of weekdays, there appeared to be no significant differences between the two groups in terms of either PA (in terms of total PA, LPA, MVPA, and daily steps) and SB patterns.

### 3.4. Total Amounts and Patterns of Physical Activity and Sedentary Behavior on Weekend

Table 4 shows the intensity-specific PA and SB patterns on the weekend. The results showed that the two groups had a significant difference only in the duration of sedentary bouts (39.8 ± 4.5 vs. 45.1 ± 5.2, *p* = 0.045). The participants attending the SSCW had a lower mean duration of sedentary bouts than those not attending the SSCW. There were no significant differences between the two groups in terms of total PA, LPA, MVPA, daily steps, total sedentary time and sedentary bout time between the two groups.

## 4. Discussion

To the best of our knowledge, this is the first study to compare the objectively-measured PA and SB of adults with PWS attending and not attending an SSCW (that was specially designed for adults with PWS) in the context of Taiwan. The key finding of this study is that the adults with PWS who attended the SSCW were more MVPA and daily step counts than those who did not during 7-day period. By stratifying daytime and nighttime on weekdays, the adults with PWS who attended the SSCW engaged in higher levels of PA (in terms of total PA, MVPA, and daily steps) and lower levels of total sedentary time than those who did not during the daytime (9:00–18:00) on weekday. However, no significant differences between the two groups in terms of either PA and SB patterns, during the nighttime. These results may have great implications for policy makers and healthcare providers insofar as they indicate that SSCW can play a key role for adults with PWS in terms of leading physically active and avoid too much sedentary time in their lifestyles.

Given that few studies have compared the PA and SB patterns of PWS patients who regularly attend and do not regularly attend a specific community center. Our novel findings showed that the adults with PWS who attending SSCW engaged sufficient MVPA (i.e. the recommendation of ≥30 daily MVPA time/day) than those who did not attend SSCW. The possible explanation is that the SSCW provides scheduled and structured classes involving PA for PWS patients during the daytime on weekdays. For example, the SSCW arranged stretching exercises and specially designed body action class for the attendees. Specifically, every attendee of the SSCW was asked to jog on a treadmill for 20–30 min once or twice a day at the SSCW. This could have been the reason why the adults with PWS who attended the SSCW engaged in a significantly higher level of MVPA (29.9 min/day vs. 13.1 min/day) during the daytime on weekdays than those who did not. Furthermore, the adults with PWS who attended the SSCW needed to regularly travel back and forth from home to the SSCW, which may also have led them to accumulate more time engaged in transport-related PA. In addition, the SSCW also requested that all the attendees take part in house cleaning, dish washing, furniture setting, soap/detergent/anti-mosquito liquid making, and various forms of skills training, all of which may have taken up time that they might otherwise have spent being sedentary during the daytime on weekdays. Conversely, the PWS patients who did not attend the SSCW may have had fewer opportunities to engage regularly structured and supervised activities in their daily life. A previous study indicated that without supervision, PWS patients may tend to have an physical inactive and prolonged sedentary time due to their low muscle tone and lack of coordination [19]. According to the self-reported daily activity forms (data not shown), the adults with PWS who did not attend the SSCW reported that they had no regular exercise or just occasionally took a walk over the 7-day period. Meanwhile, those who attended the SSCW also did not have structured PA during the nighttime on weekdays and on the weekend. Thus, it is speculated that without supervision during the daytime on weekdays, the adults with PWS who did not attend the SSCW were less physically active and more sedentary than those who attended the SSCW. The important role of the SSWC during the daytime on weekdays may also explain the interesting finding of this study that there was no difference in the parameters of PA and SB between the two groups during the nighttime on weekdays or on the weekend, with the exception of a difference in sedentary bouts on the weekend.

A number of studies have indicated that PWS patients engage in less PA than age-matched people without PWS, both among children and adults [19,20,21]. In particular, PWS patients have been found to engage in less lifestyle-related PA [21,22] and a lack of spontaneous PA [23]. In Nordstrom’s study [22], the participants with PWS (average age: 28.1 years) spent 26.2 min/day engaged in MVPA. In our study, meanwhile, the participants who attended the SSCW spent more time engaged in MVPA (49.7 min/day) than the participants in Nordstrom’s study [22], whereas those who did not attend the SSCW engaged in less MVPA (22.6 min/day) compared with those in the previous study [22]. This could also be explained by the supportive role of the structured and scheduled courses arranged by the SSCW.

Concerning LPA, previous studies showed individuals of PWS have less LPA than normal obese age-matched people [21] and other mental disorder people like Down syndrome and Williams syndrome [22]. Woods and colleges [12] indicated that, to make individual with PWS more LPA and limiting sedentary time is important. Based on a recently published review indicated that lower intensity PA or some PA is better than no PA for the health benefits in disabled adults [24]. In our study, there were no significant differences between participants who attended and did not attend the SSCW in the duration of engaging in LPA in a 7-days period, weekdays-daytime weekdays-nighttime, and weekend. Even though participants those who attended the SSCW engaging in more MVPA during weekdays-daytime than those who did not because of structured exercise, more LPA engagement was still our goal.

Therefore, SSCW should re-arrange their schedule and activities to decrease sedentary time and encourage more LPA. In Nordstrøm’s study, [22], it revealed that PA engagement of their participants (Down Syndrome, William’s syndrome, and Prader-Willi syndrome, not especially indicated PWS) did not relate to the level of support they received. Those participants who living in supported community settings had the same overall PA and PA intensities as those who lived with parents. However, our study showed the SSCW, a weekdays-daytime community center, successfully contributed to more MVPA engagement of its attendee. More research should be done to clarify that if community settings, either group homes or daycare centers, could make adults with PWS engage in more PA, as well as the impact on their body fat. Several limitations of this study should be acknowledged. First, the sample size of the study was quite small. However, our participants (18 adults with PWS) were 16.8% of all the adults with PWS in Taiwan and accounted a large proportion (40.9%) of the adults with PWS living in northern Taiwan. Second, the only significantly different demographic characteristic of the participants in the two groups in our study was age, with the adults with PWS who attended the SSCW being older than those who did not. Since PA levels generally decline with age, our findings showed that the adults with PWS who attended the SSCW were still more active than the adults with PWS who did not attend the SSCW, in spite of the participants with higher mean age of the SSCW attending group. The results thus support the important role of SSCWs in helping PWS patients to have active lifestyles. Further, more evidence on the frequency, intensity and time of PA improves the health of adults with PWS are needed [24]. Finally, the mental status of the participants was not considered due to difficulty in obtaining their medical record.

## 5. Conclusions

Our findings provide preliminary evidence regarding the important role of SSCWs, community centers with well-arranged daily schedules and proper classes during the daytime on weekdays, for adults with PWS in terms of having physically active lifestyles. Health policies or programs aiming to promote PA among adults with PWS could either consider providing structured programs or courses in a community center, or instructing the caregivers of adults with PWS to arrange their schedule towards structured programs or courses.

## Figures and Tables

**Table 1 ijerph-19-09013-t001:** Characteristic of participants.

Variables	Attending SSCW (*n* = 10)	Not Attending SSCW (*n* = 8)	*p*-Value
Age, M ± SD	28.4 ± 6.0	22.6 ± 3.9	0.032 *
Sex (%) ^a^			0.916
Male	60%	62.5%	
Female	40%	37.5%	
Height (cm), M ± SD	156.2 ± 7.8	156.9 ± 16.1	0.916
Weight (kg), M ± SD	68.5 ± 16.7	79.3 ± 15.4	0.179
BMI (kg/m^2^), M ± SD	28.2 ± 7.0	31.8 ± 6.7	0.285

BMI: body mass index; M: mean; SD: standard deviation. ^a^ chi-square analysis. * *p* < 0.05.

**Table 2 ijerph-19-09013-t002:** Total Amounts and Patterns of Physical Activity and Sedentary Behavior Patterns during 7-day period in Prader-Willi syndrome adults with and without attending SSCW.

Variables	Attending SSCW	Not Attending SSCW	*t*	*p*-Value
	M ± SD	M ± SD		
Average wear time, h/day	13.7 ± 1.0	14.7 ± 2.9	−0.97	0.35
Total PA, min/day	326.9 ± 63.7	270.2 ± 57.1	1.99	0.07
LPA, min/day	277.2 ± 50.4	247.7 ± 45.2	1.31	0.21
MVPA, min/day	49.7 ± 29.6	22.6 ± 15.9	2.33	0.03 *
Daily steps, count/day	9483.0 ± 3052.2	6260.4 ± 2733.4	2.33	0.03 *
Total sedentary time, h/day	8.2 ± 0.9	10.2 ± 3.2	−1.79	0.09
Sedentary bout duration, min/day	157.6 ± 41.8	243.1 ± 254.5	−0.94	0.38
Sedentary bout times/day	3.7 ± 1.0	3.5 ± 1.4	0.39	0.70

* *p* < 0.05; PA: physical activity; LPA: light physical activity; MVPA: moderate-to-vigorous physical activity; M: mean; SD: standard deviation.

**Table 3 ijerph-19-09013-t003:** Total amounts and patterns of physical activity and sedentary behavior in Prader-Willi syndrome adults with and without attending SSCW during the daytime and nighttime on weekdays.

Variables	Weekdays-Daytime				Weekdays-Nighttime			
	Attending SSCW	Not Attending SSCW	*t*	*p*-Value	Attending SSCW	Not Attending SSCW	*t*	*p*-Value
	M ± SD	M ± SD			M ± SD	M ± SD		
Average wear time, h/day	8.4 ± 0.4	8.5 ± 0.4	−0.43	0.68	3.5 ± 0.8	3.6 ± 0.7	−0.23	0.82
Total PA, min/day	209.5 ± 31.9	167.8 ± 29.3	2.85	0.01 *	65.5 ± 33.4	66.3 ± 20.1	−0.06	0.95
LPA, min/day	179.6 ± 29.0	154.7 ± 28.4	1.83	0.09	55.6 ± 22.9	60.8 ± 17.4	−0.53	0.60
MVPA, min/day	29.9 ± 18.2	13.1 ± 5.8	2.50	0.02 *	9.9 ± 11.1	5.5 ± 5.7	1.00	0.33
Daily steps, count/day	6396.0 ± 2492.2	4047.6 ± 1650.2	2.29	0.04 *	1492.3 ± 1386.8	1256.7 ± 747.3	0.43	0.67
Total sedentary time, h/day	4.9 ± 0.5	5.7 ± 0.3	−3.79	0.002 *	2.4 ± 0.5	2.5 ± 0.5	−0.32	0.76
Sedentary bout duration, min/day	91.6 ± 26.2	100.9 ± 58.5	−0.42	0.69	53.6 ± 27.4	48.9 ± 30.0	0.34	0.74
Sedentary bout times/day	2.1 ± 0.6	2.01 ± 1.0	0.19	0.86	1.2 ± 0.6	1.0 ± 0.6	0.64	0.53

* *p* < 0.05; PA: physical activity; LPA: light physical activity; MVPA: moderate-to-vigorous physical activity; M: mean; SD: standard deviation.

**Table 4 ijerph-19-09013-t004:** Total amounts and patterns of physical activity and sedentary behavior in Prader-Willi syndrome adults with and without attending SSCW on weekend.

Variables	Attending SSCW (*n* = 9) ^a^	Not Attending SSCW (*n* = 7) ^b^	*t*	*p*-Value
	M ± SD	M ± SD		
Average wear time, h/day	13.7 ± 1.7	13.7 ± 1.6	−0.03	0.98
Total PA, min/day	323.7 ± 102.6	280.9 ± 59.1	0.98	0.34
LPA, min/day	272.2 ± 79.2	255.9 ± 46.1	0.48	0.64
MVPA, min/day	51.5 ± 40.5	25.0 ± 24.7	1.52	0.15
Daily steps, count/day	8888.5 ± 3595.3	6843.0 ± 2976.5	1.21	0.25
Total sedentary time, h/day	8.3 ± 1.5	9.0 ± 1.7	−0.95	0.36
Sedentary bout duration, min/day	39.8 ± 4.5	45.1 ± 5.2	−2.20	0.04 *
Sedentary bout times/day	4.2 ± 0.9	4.1 ± 2.4	0.10	0.92

* *p* < 0.05; PA: physical activity; LPA: light physical activity; MVPA: moderate-to-vigorous physical activity; M: mean; SD: standard deviation.; ^a^ There was no accelerometer data on the weekend from one of the participating adults with PWS attending the SSCW; thus, his/her data was excluded from the analysis.; ^b^ There is was no accelerometer data on the weekend from one of the participating adults with PWS not attending the SSCW; thus, his/her data was excluded from the analysis.

## Data Availability

Data used in this study are available upon reasonable requests.

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
