# Peer review of "Accelerometer-Measured Physical Activity and Sedentary Behavior of Adults with Prader-Willi Syndrome Attending and Not Attending a Small-Scale Community Workshop"

_ijerph, 2022, doi:10.3390/ijerph19159013_

Round 1

Reviewer 1 Report

Ming-Ju Wu et al. are presenting a study of Prader-Willi Syndrome patients who do or do not attend a regular supervised community program. Using several parameters, they compare the physical activity of the two groups and find that participation in the community program leads to significantly higher physical activity. Though the effects are moderate the authors conclude they likely have long-term health benefits for the patients.

The study design is straightforward, the article is clearly written, and the procedures are described in suitable detail. The topic warrants sufficient general interest.

Minor points:

·       line 137, line 155, table 1:  as the study is about biology and not the social roles of men and women, I find “sex” is the correct word to use, not “gender”.

·       line 176, line 217: replace “did not attending” with “did not attend”

·       line 259: Replace “Based on recently review” with “Based on a recently published review

·       line 261: Replace “between participants those who attended and not attended” with “between participants who attended and did not attend

·       line 267: Please rephrase, not clear: “Therefore, SSCW should re-arrange their schedule and activities to decrease sedentary time then make participants who attended the SSCW will engage in more LPA

·       line 271:  replace ”did not related” with “did not relate

·       line 274: replace “More researches should be done” with “More research should be done

·       line 286: Replace “Further, the evidence on (…) are needed” with “Further, more evidence on (…) are needed

Author Response

Responses to the Reviewer 1:

Ming-Ju Wu et al. are presenting a study of Prader-Willi Syndrome patients who do or do not attend a regular supervised community program. Using several parameters, they compare the physical activity of the two groups and find that participation in the community program leads to significantly higher physical activity. Though the effects are moderate the authors conclude they likely have long-term health benefits for the patients.

The study design is straightforward, the article is clearly written, and the procedures are described in suitable detail. The topic warrants sufficient general interest.

Response: Thank you very much for your positive comments.

Minor points:

Query 1: line 137, line 155, table 1:  as the study is about biology and not the social roles of men and women, I find “sex” is the correct word to use, not “gender”.

Response 1: Thank you very much for your suggestions. We have revised accordingly.

Query 2: line 176, line 217: replace “did not attending” with “did not attend”

Response 2: Thank you very much for your suggestions. We have revised accordingly.

Query 3: line 259: Replace “Based on recently review” with “Based on a recently published review”

Response 3: Thank you very much for your suggestions. We have revised accordingly.

Query 4: line 261: Replace “between participants those who attended and not attended” with “between participants who attended and did not attend”

Response 4: Thank you very much for your suggestions. We have revised accordingly.

Query 5: line 267: Please rephrase, not clear: “Therefore, SSCW should re-arrange their schedule and activities to decrease sedentary time then make participants who attended the SSCW will engage in more LPA”

Response 5: Thank you very much for your comments. We have rephrased this sentence accordingly. (page 7)

Therefore, SSCW should re-arrange their schedule and activities to decrease sedentary time and encourage more LPA

Query 6: line 271:  replace ”did not related” with “did not relate”

Response 6: Thank you very much for your suggestions. We have revised accordingly.

Query 7: line 274: replace “More researches should be done” with “More research should be done”

Response 7: Thank you very much for your suggestions. We have revised accordingly.

Query 8: line 286: Replace “Further, the evidence on (…) are needed” with “Further, more evidence on (…) are needed”

Response 8: Thank you very much for your suggestions. We have revised accordingly.

Reviewer 2 Report

This is a prospective study in which the author evaluated the impact of a small-scale community workshop on physical activity among adults

with Prader-Willi Syndrome. The results showed increased activity among participants, which is excellent. Still, it would have been nice to 

evaluate the impact of increased activity, like a decrease in body weight or total body fat.

Please clarify if participants were encouraged to be more active at small-scale community workshops. If yes, then how was that done.

The follow-up duration was 6 months to 5 years; please provide the average follow-up duration in the results.

Were the anthropometric measurements done during the study and at the end of the study? If yes, then please make a table of those values.

Was there intervention done on calorie intake during this follow-up? Please clarify that in the method section since the reader will be interested in knowing that.

Author Response

Responses to the Reviewer 2:

This is a prospective study in which the author evaluated the impact of a small-scale community workshop on physical activity among adults with Prader-Willi Syndrome.

Response: Thank you very much for your comments. In fact, this study is a cross-sectional study. We have added this important information in Abstract (page 1) and Method (page 2).

This cross-sectional study aimed to compare the accelerometer-assessed physical activity (PA) and sedentary behavior (SB) of adults with Prader-Willi syndrome (PWS) attending or not attending a small-scale community workshop (SSCW).

Among these adults with PWS, a total of 44 were living in northern Taiwan and were the target population for our study. This study is a cross-sectional design.

Query 1: The results showed increased activity among participants, which is excellent. Still, it would have been nice to evaluate the impact of increased activity, like a decrease in body weight or total body fat.

Response 1: Thank you very much for your comments. First of all, yes, we found that the adults with PWS who attended the SSCW engaged in more moderate-to-vigorous PA (MVPA) and daily steps than those who did not. Second, we did not evaluate the differences in body weight or total body fat. We have suggested that future studies can further examine this issue. (page 7).

More research should be done to clarify that if community settings, either group homes or daycare centers, could make adults with PWS engage in more PA, as well as the impact on their body fat.

Query 2: Please clarify if participants were encouraged to be more active at small-scale community workshops. If yes, then how was that done.

Response 2: Thank you very much for your comments. No particular strategies were used to encourage or motivate the participants to be active at small-scale community workshop (SSCW). Instead, SSCW provides scheduled and structured classes involving PA for PWS patients, such as stretching exercises, body action class and jogging on a treadmill. We have provided this important information in the Discussion (page 6).

The possible explanation is that the SSCW provides scheduled and structured classes involving PA for PWS patients during the daytime on weekdays. For example, the SSCW arranged stretching exercises and specially designed body action class for the attendees. Specifically, every attendee of the SSCW was asked to jog on a treadmill for 20-30 minutes once or twice a day at the SSCW.

Query 3: The follow-up duration was 6 months to 5 years; please provide the average follow-up duration in the results.

Response 3: Thank you very much for your comments. As our response of Query 1, this study is a cross-sectional design, thus, it is not possible to provide average follow-up duration.

Query 4: Were the anthropometric measurements done during the study and at the end of the study? If yes, then please make a table of those values.

Response 4: Thank you very much for your comments. We only measured anthropometric outcomes one time during the study. As a result, we cannot provide the information of those values.

Query 5: Was there intervention done on calorie intake during this follow-up? Please clarify that in the method section since the reader will be interested in knowing that.

Response 5: Thank you very much for your comments. We did not do any intervention on calorie intake and thus we cannot provide such information. 
